# Influence of Technostress on Academic Performance of University Medicine Students in Peru during the COVID-19 Pandemic

**Aldo Alvarez-Risco** [1], **Shyla Del-Aguila-Arcentales** [2], **Jaime A. Yáñez** [3,4,*], **Marc A. Rosen** [5] and **Christian R. Mejia** [6]

1  Carrera de Negocios Internacionales Facultad de Ciencias Empresariales y Económicas, Universidad de Lima, Lima 15023, Peru; aralvare@ulima.edu.pe
2  Escuela Nacional de Marina Mercante "Almirante Miguel Grau", Callao 07021, Peru; sdelaguila@enamm.edu.pe
3  Carrera de Educación y Gestión del Aprendizaje, Facultad de Educación, Universidad Peruana de Ciencias Aplicadas, Lima 15023, Peru
4  Gerencia Corporativa de Asuntos Científicos y Regulatorios, Teoma Global, Lima 15073, Peru
5  Faculty of Engineering and Applied Science, University of Ontario Institute of Technology, Oshawa, ON L1G 0C5, Canada; marc.rosen@uoit.ca
6  Translational Medicine Research Centre, Universidad Norbert Wiener, Lima 15046, Peru; christian.mejia.md@gmail.com
*  Correspondence: jaimeayanez@gmail.com

**Abstract:** The current study aims to validate and apply an instrument to assess the relationship between communication overload, social overload, technostress, exhaustion and academic performance. We performed a cross-sectional, analytical study of 2286 university medical students to assess the influence of technostress as a mediator of social media overload, communication overload and mental exhaustion and its detrimental effect on the academic performance of university students in Peru during the COVID-19 pandemic. The research model was validated using partial least square structural equation modeling (PLS-SEM) to establish the influence of variables on the model. Communication and social overload were found to positively influence technostress by correlations of 0.284 and 0.557, respectively. Technostress positively influenced exhaustion by 0.898, while exhaustion negatively influenced academic performance by -0.439. Bootstrapping demonstrated that the path coefficients of the research model were statistically significant. The research outcomes may help university managers understand students' technostress and develop strategies to improve the balanced use of technology for their daily academic activities.

**Keywords:** Peru; social media overload; academic performance; university students; COVID-19; pandemic

## 1. Introduction

The COVID-19 pandemic has reached 173 million confirmed cases and over 3.7 million deaths worldwide as of 7 June 2021 [1]. Physical isolation was the main preventive measure implemented worldwide to avoid contagion [2–4], which caused multiple lifestyle changes in people. Many people have experienced the death of family and friends [5–9], which has resulted in anxiety and mental distress [10–13]. The widespread disinformation [14,15], fake news [16] and anti-vaccine comments [17,18] have caused an increase in self-medication [19], use of medicinal plants [20] and other alternative treatments [21]. Many have urged that the general state of disinformation be addressed by governmental institutions [22]. Because people wanted to stay informed, they accessed various online resources, which could contribute to communication and social overload. Due to the generalized lockdown measures around the globe, the use of information technology surged to permit telemedicine [23], telework [24,25] and online classes [26–29].

The use of technology in universities' learning processes started some years ago [30], and it has generated various educational benefits for both teachers and students [31]. However, it has also been observed to reduce engagement in collaborative learning, student and professor interaction and bidirectional debate compared to teaching in the traditional classroom [32]. Before the pandemic, it was reported that users in China dedicated 33% of their daily internet time to social networks [33]. This percentage significantly increased during the COVID-19 pandemic [34] and was accompanied by a progressive increase in the use of social networks by universities to post information about their classes and how to access virtual learning resources [35]. However, the overload that extended use of social networks can cause has increased the time students remain connected to social networks, especially since it may overlap with their academic responsibilities. This can lead to "technostress", which has been reported to affect the quality of sleep [36] and academic performance [37,38].

The objective of the current study is to assess the influence of technostress as a mediator of social media overload, communication overload and mental exhaustion and its detrimental effect on the academic performance of university students in Peru during the COVID-19 pandemic. The research may help professors, university managers and practitioners understand students' technostress and develop strategies to improve the balanced use of technology for their daily academic activities.

The content of the paper is structured as follows: Section 2 presents relevant background. The methodology, with a description of the instrument, sample and data processing, is provided in Section 3. Section 4 gives the results and outcomes according to the questionnaire applied, and these are discussed in Section 5. Conclusions with theoretical, practical and societal implications and recommendations, including potential future research, are provided in Section 6.

## 2. Theoretical Framework

Koeske and Koeske [39] proposed the stress–strain–outcome model that allows an understanding, in an articulated way, of the influence of a stressor on academic or work performance. According to this model, any environmental stimulus that can disturb the usual activities of a student is considered a stress factor. Furthermore, this model considers the negative results that these factors have on academic performance, which are amplified from the exhaustion caused by the stress factor.

The term "technostress" was used for the first time by Brod [40], and it was defined as "a modern disease of adaptation caused by an inability to cope with the new computer technologies in a healthy manner." A series of individual and organizational strategies to deal with technostress were also reported by Kupersmith [41].

Technostress can be considered the result of a higher academic/work demand to have high digital skills to carry out daily activities. As can be observed in the literature, several studies describe the influence of technostress on the performance of workers [42–48]. Currently, it is common for students to extensively use the internet for both their academic and personal activities, with social networks taking a great deal of their time during their day. It has been reported that technostress can influence the academic performance of students [37,49] as well as teachers [50–52]. During the COVID-19 pandemic, people were suddenly forced into a remote working situation [53] that often led to a work–home conflict [54].

It has been reported that the pandemic has generated various alterations in the work–family interface [55], evidenced by increased workaholism and anxiety [56] as triggers of technostress. Technostress has already been reported among university teachers in Egypt [57] and Spain [58] during the COVID-19 pandemic; however, to the best of our knowledge, no studies have been reported on technostress in university students.

## 3. Approach

### 3.1. Research Model

Figure 1 shows the research model detailing the relationship between the study variables, which was adapted from Shi et al. [59]. The model includes education development support, conceptual development support and country support through entrepreneurial self-efficacy and green entrepreneurial intention. In Figure 1, the circles represent each variable of the study.

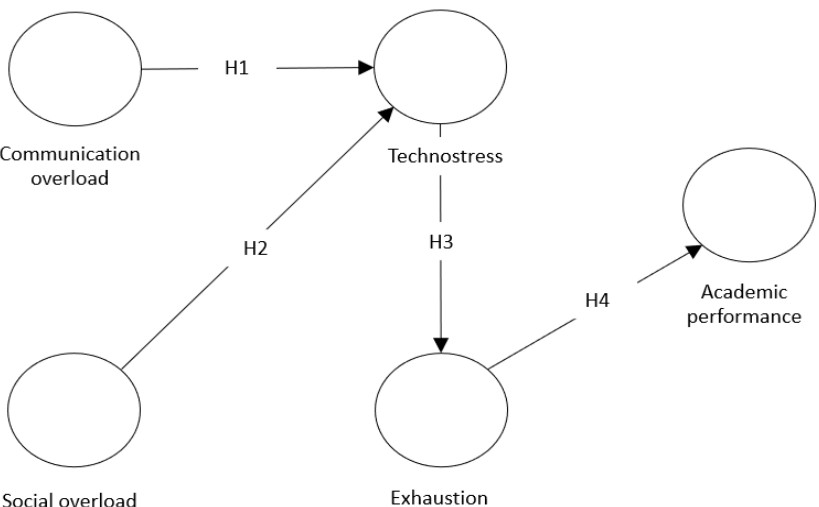

**Figure 1.** Research model.

### 3.2. Development of Hypothesis

It is hypothesized that communication overload and social overload lead to technostress [59–62], which causes exhaustion and negatively influences academic performance [59,63,64]. The following hypotheses are proposed to describe the complete path to connect the variables:

**Hypothesis 1 (H1).** *Communication overload has a positive influence on technostress.*

**Hypothesis 2 (H2).** *Social overload has a positive influence on technostress.*

**Hypothesis 3 (H3).** *Technostress has a positive influence on exhaustion.*

**Hypothesis 4 (H4).** *Exhaustion has a negative influence on academic performance.*

## 4. Methodology

The methodology used in the current study includes an observational study with both a descriptive and an inferential design.

### 4.1. Data Sample

Data on university medical students from the 25 regions of Peru were collected through non-probabilistic sampling. Participants completed an online questionnaire administered 11 July–28 August 2020. We distributed surveys to 2800 students, and 2286 questionnaires were completed and submitted.

### 4.2. Data Collection and Instrument

The questionnaire (Annex 1) consists of two sections. The first part collects demographic information from university students, while the second part consists of questions based on the instruments developed to assess communication overload [65], social overload [66], technostress [67], exhaustion [66] and academic performance [68]. The original

items were translated into Spanish and adapted linguistically. The section related to communication overload consists of three items, social overload consists of five items, technostress consists of four items, exhaustion consists of three items and academic performance consists of four items. All the items were assessed through a Likert-type scale of five response options (from 1 = completely disagree to 5 = completely agree). The origins of the scales and items used in the current study are presented in Table 1.

**Table 1.** Variables and items.

| Variable | Item | Reference |
|---|---|---|
| Communication overload | I feel that I generally receive too many notifications on new postings, push messages, and news feeds, among others from social media as I perform other tasks<br>I often feel overloaded with social media communication<br>I receive more communication messages and news from friends on social media than I can process | Cho et al. (2011) |
| Social overload | I take too much care of the well-being of my friends on social media<br>I deal with my friends' problems on social media too much<br>My sense of responsibility for how much fun my friends have on social media is too strong<br>I care for my friends on social media too often<br>I pay too much attention to my friends' posts on social media | Maier et al. (2015) |
| Technostress | I am forced by social media to live very tight time schedules<br>I am forced to change habits to adapt to new developments on social media<br>I have to sacrifice my personal time to keep current on new social media updates<br>I feel my personal life is being invaded by social media | Ragu-Nathan et al. (2008) |
| Exhaustion | I feel drained from activities that require me to use social media<br>I feel tired from my social media activities<br>Using social media is a strain for me | Maier et al. (2015) |
| Academic performance | I am confident about the adequacy of my academic skills and abilities<br>I feel competent conducting my course assignments<br>I have learned how to successfully perform my coursework in an efficient manner<br>I have performed academically as well as I anticipated I would | Yu et al. (2010) |

*4.3. Data Analysis*

The data collected were tabulated and analyzed using the statistical programs SPSS version 26. The data quality was evaluated to ensure the originality of the source and to delete incomplete questionnaires. Each subscale's internal consistency was evaluated by Cronbach's alpha reliability coefficient [69]. The questionnaire was validated using partial least square structural equation modeling (PLS-SEM), which was performed in SmartPLS statistical package version 3.3.2. PLS-SEM was used to determine the construct and discriminant validity, as well as the internal consistency through composite reliability [70]. The reliability of each indicator was assessed by examining its loads and their dimension, with load values higher than 0.50 considered reliable. A good fit for the average extracted variance values was established at higher than 50%. Finally, the questionnaire's discriminant validity was established by applying the Fornell–Larcker criterion [71], which indicates that the square root of variance extracted must be greater than the correlations presented with the rest of the subscales. To verify that the influence between the variables in the research model was statistically significant, the non-parametric technique of bootstrapping was used [72]. For this, a resample strategy (5000 resamples) was applied to test if the path coefficients (beta) were significant.

*4.4. Ethical Aspects*

The survey was approved by the Universidad Privada Antenor Orrego ethics committee (#0239-2020-UPAO). The participants remained anonymous and had the option to finish the survey at any time, and their information was kept confidential. All the survey participants were well versed on the study intentions and were required to consent before enrollment. The participants were not involved in any of the planning, execution or reporting stages of the study.

## 5. Results

The survey was completed by 2286 participants from 43 cities in the 25 regions of Peru. The majority of the respondents were men (1292 (56.4%)), ranging between 18 and 26 years of age.

*5.1. Reliability*

The scales for communication overload, social overload, technostress, exhaustion and academic performance exhibited reliability coefficients (Cronbach's Alpha) higher than the expected minimum of 0.5 in the exploratory analysis using partial least square structural equation modeling (PLS-SEM) (see Table 2).

**Table 2.** Analysis of internal consistency using partial least square structural equation modeling (PLS-SEM).

| Scale | N° of Items | Cronbach's Alpha | Range of Relations Item–Scale |
|---|---|---|---|
| Communication overload | 3 | 0.919 | 0.907–0.949 |
| Social overload | 5 | 0.927 | 0.834–0.922 |
| Technostress | 4 | 0.945 | 0.922–0.931 |
| Exhaustion | 3 | 0.942 | 0.935–0.955 |
| Academic performance | 4 | 0.919 | 0.865–0.926 |

Sample: 2286 questionnaires completed by university students.

To verify the validity of the instrument, the research model was analyzed using PLS-SEM, which included the reliability analysis of each indicator, the internal consistency of each dimension (composite reliability), the analysis of the average variance extracted and the discriminant validity.

*5.2. Composite Reliability*

An acceptable level of composite reliability must be greater than 0.70 [69]. The coefficients of reliability composed of the different subscales of the instrument ranged between 0.816 and 0.900 (Table 3). Overall, the values obtained for the three subscales confirm the reliability of the questionnaire.

**Table 3.** Construct validity of the items using partial least square structural equation modeling (PLS-SEM).

| Scale Items | Composite Reliability | Extracted Variance |
|---|---|---|
| **Communication overload** | | |
| I feel that I generally receive too many notifications on new postings, push messages and news feeds, among others from social media as I perform other tasks. | | |
| I often feel overloaded with social media communication. | 0.849 | 0.761 |
| I receive more communication messages and news from friends on social media than I can process. | | |
| I feel that I generally receive too many notifications on new postings, push messages and news feeds, among others from social media as I perform other tasks. | | |

**Table 3.** *Cont.*

| Scale Items | Composite Reliability | Extracted Variance |
|---|---|---|
| **Social overload**<br>I take too much care of the well-being of my friends on social media<br>I deal with my friends' problems on social media too much<br>My sense of responsibility for how much fun my friends have on social media is too strong<br>I care for my friends on social media too often<br>I pay too much attention to my friends' posts on social media | 0.845 | 0.674 |
| **Technostress**<br>I am forced by social media to live very tight time schedules<br>I am forced to change habits to adapt to new developments on social media<br>I have to sacrifice my personal time to keep current on new social media updates<br>I feel my personal life is being invaded by social media | 0.861 | 0.759 |
| **Exhaustion**<br>I feel drained from activities that require me to use social media<br>I feel tired from my social media activities<br>Using social media is a strain for me | 0.862 | 0.795 |
| **Academic performance**<br>I am confident about the adequacy of my academic skills and abilities<br>I feel competent conducting my course assignments<br>I have learned how to successfully perform my coursework in an efficient manner<br>I have performed academically as well as I anticipated I would | 0.842 | 0.703 |

Sample: 2286 questionnaires completed by university students.

*5.3. Discriminant Validity*

The Fornell–Larcker criterion was used to establish the subscales' discriminant validity [71].

The assessment of discriminant validity involves latent variables for the prevention of multicollinearity issues. Table 4 shows compliance with this criterion in all subscales (diagonals between parentheses), demonstrating the discriminant validity of the instrument analyzed.

**Table 4.** Discriminant validity of subscales using the Fornell–Larcker criterion.

| Scale | Academic Performance | Communication Overload | Exhaustion | Social Overload | Technostress |
|---|---|---|---|---|---|
| Academic performance | (0.896) | | | | |
| Communication overload | −0.382 | (0.928) | | | |
| Exhaustion | −0.439 | 0.670 | (0.946) | | |
| Social overload | −0.453 | 0.671 | 0.697 | (0.880) | |
| Technostress | −0.464 | 0.658 | 0.898 | 0.748 | 0.927 |

Sample: 2286 questionnaires completed by university students.

*5.4. Bootstrapping*

The criterion used was to compare the original value obtained from the model with the average obtained from carrying out 5000 resamples. The original value is expected to be very similar to the average value obtained [72], which allows us to assert if the model is significant. According to Table 5, all relations are significant (p values <0.01) except the relation between communication overload and technostress, which must be evaluated and verified in future studies.

**Table 5.** Significance of trajectory coefficients (beta).

| Scale | Original Sample | Mean Sample | Standard Deviation | t-Statistic | *p* |
|---|---|---|---|---|---|
| Communication overload → Technostress | 0.284 | 0.281 | 0.143 | 1.989 | 0.047 |
| Exhaustion → Academic performance | −0.439 | −0.449 | 0.089 | 4.912 | 0.000 |
| Social overload → Technostress | 0.557 | 0.561 | 0.124 | 4.511 | 0.000 |
| Technostress → Exhaustion | 0.898 | 0.898 | 0.027 | 33.126 | 0.000 |

Bootstrapping technique (5000 times) using Smart PLS. *p* value <0.01. Source: 2286 questionnaires completed by university students.

*5.5. Model Evaluation*

Figure 2 shows the evaluation of the research model. It can be observed that communication overload and social overload positively influence technostress. Simultaneously, it was shown that technostress positively influences exhaustion. Finally, exhaustion has a negative influence on students' academic performance.

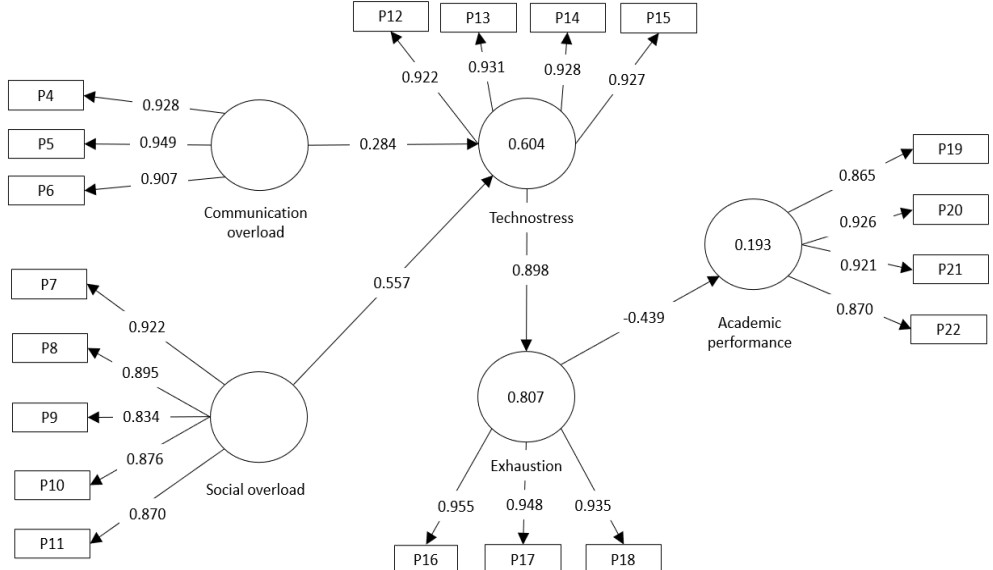

**Figure 2.** Research model testing.

*5.6. Test of Hypothesis*

**Hypothesis 5 (H5).** *Communication overload has a positive influence on technostress.*

Communication overload has a positive influence of 0.284 on technostress.

**Hypothesis 6 (H6).** *Social overload has a positive influence on technostress.*

Social overload has a positive influence of 0.557 on technostress. Social overload together with communication overload explains 60.4% of technostress.

**Hypothesis 7 (H7).** *Technostress has a positive influence on exhaustion.*

Technostress has a positive influence of 0.898 on exhaustion. Technostress explains 80.7% of exhaustion.

**Hypothesis 8 (H8).** *Exhaustion has a negative influence on academic performance.*

Exhaustion has a negative influence of −0.439 on academic performance. Exhaustion explains 19.3% of academic performance.

## 6. Discussion

The influence of technostress on the productivity of information and communication technology (ICT) users and their satisfaction has been evaluated [73,74]. It was observed that workers in more centralized companies and with a high demand for innovation present more technostress [75]. It has been reported that older users have more difficulty in adjusting to the new technologies, which has become more evident during the COVID-19 pandemic [76].

The pandemic has accentuated the work–home conflict [54], a situation where working from home can lead to interrupting work progress to attend to home affairs and, similarly, to interrupting household chores to complete work obligations. Various mitigation measures have been proposed for organizations to reduce technostress, which include more training for users of technological systems [77]. However, during the COVID-19 pandemic, we were suddenly forced into a remote working situation [53]. It has been reported that the pandemic has generated various alterations in the work–family interface [55], leading to workaholism [53] and anxiety [56] as triggers of technostress. More technostress has been observed in 2020 compared to past years, suggesting support is needed to overcome technostress and carry out academic work [78]. Finally, workplace monitoring and technostress issues are expected to increase and become prominent due to a continually increasing digital presence [78].

When the traditional telephone was a fixed device in the home or office, there was a degree of freedom that has been lost since smartphones have become almost omnipresent [79]. The use of mobile phones has caused people to be able to be localized anytime and everywhere via various communication apps and social networks, which has led to a feeling of urgency to respond, even at the expense of interrupting work and academic and/or social life, which can lead to a high level of communication overload [80,81]. This has increased during the COVID-19 pandemic because social isolation has increased the need for virtual communication [56]. In recent years, social networks have had a leading role in people's lives due to the immense amount of information and the speed at which the information is updated. Students typically combine their academic time attending class and doing homework with constant attention to social networks in order to maintain their presence in the cyber community [16]. Due to the COVID-19 lockdown and social isolation measures, an increase in social network posts has been observed as a response to maintain social connection with other people through likes and comments. Then, the free time used for social networks began to be combined with the need to be online to attend other activities than mere leisure. In students, this has created conditions that facilitate the development of technostress due to the substantial virtual overload that they experience [36].

Communication and social overload can cause technostress, which generates fatigue and physical and emotional exhaustion leading to the desire to disconnect from technology [60]. The problem with exhaustion is that it escapes the user's control, causing repercussions in people's daily lives, such as sleeping and concentration problems, among others [63]. Thus, workers with technostress manifest exhaustion that affects productivity and job performance [64]. The same occurs with students who, due to communication and social overload, develop technostress and exhaustion that affect their capacity to complete homework and study for tests [59].

As mentioned earlier, studies have not reported on the effects of technostress on university students during the pandemic, but they have examined its effect on university teachers [57]. The present results show the influence of communication overload (0.284) on technostress, which could be related to the multiple instructions and coordination that students receive over the internet. It has been reported that communication overload can lower productivity [61] and affect students' academic performance [59,62]. Our results show that social overload (0.557) influences technostress, possibly related to the more frequent interaction on social networks and the virtual platforms of universities. The virtual platforms tend to have multiple messages and publications, often making it difficult to manage and attend to so much information, thus contributing to the technostress

reported. This has been previously reported where the effect of social media information increased the technostress in university students [37]. Our results show that technostress (0.898) causes exhaustion, and exhaustion negatively influences academic performance (−0.439). Similar results were observed in university students in China, who exhibited exhaustion caused by technostress [59]; it was also determined that social support could help reduce the extent of exhaustion [82].

The present study provides a valuable contribution since it allows the validation and application of an instrument to assess the influence of communication and social media overload on academic performance in university medical students in Peru during the COVID-19 pandemic. The results show that the questionnaire used is valid and reliable. Likewise, due to bootstrapping data, it can also be inferred that the model is significant. Our results show the route to how communication overload, social overload and technostress lead to exhaustion, which ultimately affects academic performance. During the COVID-19 pandemic, students have been forced to study, complete homework and attend their social networks from home. Our results are similar to previous reports, where the extensive use of mobile devices in university students contributes to the development of technostress and negatively affects academic performance [38]. Similarly, it was reported that information overload, communication overload and social overload influence the development of technostress and exhaustion, negatively affecting academic performance in Chinese university students [59]. Furthermore, the influence of technostress on academic performance has been reported for Indian university students [83].

The data of our study are relevant not only for universities but also for companies, allowing them to be aware that digital information overload can cause technostress in their employees. At the same time, technostress can cause exhaustion and generate a negative influence on job performance. Therefore, the present authors recommend that communication during telework be balanced since an overload of emails can cause the development of technostress. Finally, and perhaps not surprisingly, companies should be aware that students subjected to technostress during their academic training can easily succumb to the same problem when they are part of the workforce.

*Limitations*

Our data were collected in Peru, a country with limited access to the internet, and it remains unclear whether the severeness of technostress is similar in other countries that might have more access to social media or the internet. We surveyed university medical students, but technostress also needs to be evaluated in other professional careers and in other groups of people. It remains to be determined which is the social network that demands the most time for students. In the future, it would also be useful to obtain information from students concerning possible strategies that they have used to achieve a balance between attention to social networks and the fulfillment of academic tasks. Further effort to determine efficient procedures to generate a reasonable and efficient use of the internet, especially social networks, is warranted.

## 7. Conclusions

The COVID-19 pandemic has changed many aspects of people's lives and accentuated work–home conflict, which can lead to an increase in the incidence of technostress. The results of this investigation show that communication and social overload positively influence the development of technostress and exhaustion, and exhaustion negatively influences academic performance. The results may help university managers to understand students' technostress and develop strategies to improve the balanced use of technology in their daily academic activities. It is important to understand the factors that influence technostress and academic performance during the virtual learning modality, as this could help future research to determine the necessary steps to take during the return to regular, in-person learning after the COVID-19 pandemic.

**Author Contributions:** Conceptualization, A.A.-R., S.D.-A.-A., J.A.Y. and C.R.M.; methodology, A.A.-R., J.A.Y. and C.R.M.; validation, A.A.-R., S.D.-A.-A., J.A.Y. and C.R.M.; formal analysis, A.A-R; investigation, C.R.M.; data curation, S.D.-A.-A. and A.A.-R.; writing—original draft preparation, A.A.-R., S.D.-A.-A., J.A.Y., M.A.R. and C.R.M.; writing—review and editing, A.A.-R., S.D.-A.-A., J.A.Y., M.A.R. and C.R.M.; visualization, A.A.-R., J.A.Y., M.A.R. and C.R.M. All authors have read and agreed to the published version of the manuscript.

**Funding:** The authors financed this work.

**Institutional Review Board Statement:** The survey was approved by the Universidad Privada Antenor Orrego ethics committee (#0239-2020-UPAO).

**Informed Consent Statement:** All the survey participants were well versed on the study intentions and were required to consent before enrollment.

**Data Availability Statement:** The data presented in this study are available on request from the corresponding author.

**Conflicts of Interest:** The authors declare no conflict of interest.

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
