# Peer review of "Influence of Technostress on Academic Performance of University Medicine Students in Peru during the COVID-19 Pandemic"

_sustainability, doi:10.3390/su13168949_

Round 1

Reviewer 1 Report

The aim of this paper is to perform a cross-sectional, analytical, study of 2286 university students to assess the influence of technostress as a mediator of social media overload, communication overload and mental exhaustion and its detrimental effect on the academic performance of university students in Peru during the COVID-19 pandemic. Overall, the concept of this paper is promising, the structure of the paper is correct and it offers some good arguments.

The main strengths of this paper are the following:

  • The title accurately reflects the content of this study.
  • The tables and figures are presented clearly and analyzed appropriately.
  • References cover seminal literature.
  • The paper is well-written.

First of all, the abstract of the paper is not complete and stand-alone. Authors provided some details about the methodology used and presented the main findings of their analysis. However, they should mention the objective as well as the practical implication of this research. Furthermore, the authors should highlight the motivation and the practical contribution of the paper.

The Introduction is focused. Authors used the traditional structure in order to justify the research gap and the motivation as well as the value of this paper. Authors presented the motivation of the paper and discussed about the main findings. However, authors should add the contribution of this paper to academics and practitioners.

The paper demonstrated an adequate understanding of the relevant literature in the field and cited an appropriate range of literature sources. Authors analyzed the findings and research gaps from previous researchers and clearly identified hypotheses. However, figure 1 should be placed at the end of the second section. Authors are recommended to use more references in order to discuss about the hypotheses defined that should also be placed at the end of the second section.

Section 4 presents the research methodology which is clear. The research on which the paper is based is well designed and the methods that have been employed are appropriate. This section is designed based on the existing literature. The methodology is sound and the statistical analysis is relevant. However, authors should refer the number of questionnaires that have been distributed to students.

The findings are a good basis for discussion and the results help unleash new insights in this discipline. Authors compared these results with the findings from previous similar studies and conclusions are supported by the data. Authors presented implications and clarified limitations and suggestions for future research.

Author Response

Reviewer #1

We thank the reviewer for the positive review of our manuscript. The reviewer has made some critical and insightful comments that have definitely improved the final version. We have carefully amended the paper as suggested by the reviewer.

Comments

The aim of this paper is to perform a cross-sectional, analytical, study of 2286 university students to assess the influence of technostress as a mediator of social media overload, communication overload and mental exhaustion and its detrimental effect on the academic performance of university students in Peru during the COVID-19 pandemic. Overall, the concept of this paper is promising, the structure of the paper is correct and it offers some good arguments.

The main strengths of this paper are the following:

  • The title accurately reflects the content of this study.
  • The tables and figures are presented clearly and analyzed appropriately.
  • References cover seminal literature.
  • The paper is well-written.
  1. First of all, the abstract of the paper is not complete and stand-alone. Authors provided some details about the methodology used and presented the main findings of their analysis. However, they should mention the objective as well as the practical implication of this research. Furthermore, the authors should highlight the motivation and the practical contribution of the paper.

We thank the reviewer for this comment, it has helped us to improve the revised manuscript. We have added to the abstract the following statement to include the objective of the study: “The current study aims to validate and apply an instrument to assess the relationship between communication overload, social overload, technostress, exhaustion, and academic performance.” The practical contribution was already part of the abstract, on the final sentence: “The research outcomes may help university managers understand students' technostress and develop strategies to improve the balanced use of technology for their daily academic activities.”

  1. The Introduction is focused. Authors used the traditional structure in order to justify the research gap and the motivation as well as the value of this paper. Authors presented the motivation of the paper and discussed about the main findings. However, authors should add the contribution of this paper to academics and practitioners.

We thank the reviewer for this comment, we have included the following statement in the introduction section: “The research may help professors, university managers and practitioners understand students' technostress and develop strategies to improve the balanced use of technology for their daily academic activities.”

  1. The paper demonstrated an adequate understanding of the relevant literature in the field and cited an appropriate range of literature sources. Authors analyzed the findings and research gaps from previous researchers and clearly identified hypotheses. However, figure 1 should be placed at the end of the second section. Authors are recommended to use more references in order to discuss about the hypotheses defined that should also be placed at the end of the second section.

We thank the reviewer for the positive comments about our research, we have moved Figure 1 after section 3.2. Development of hypothesis. We have also added references in section 3.2.

  1. Section 4 presents the research methodology which is clear. The research on which the paper is based is well designed and the methods that have been employed are appropriate. This section is designed based on the existing literature. The methodology is sound and the statistical analysis is relevant. However, authors should refer the number of questionnaires that have been distributed to students.

We thank the reviewer for the positive comments about our research, we have included the following sentence in section 4.1: We distributed surveys to 2800 students.

  1. The findings are a good basis for discussion and the results help unleash new insights in this discipline. Authors compared these results with the findings from previous similar studies and conclusions are supported by the data. Authors presented implications and clarified limitations and suggestions for future research.

We thank the reviewer for the positive comments about our research.

Reviewer 2 Report

It is an interesting article for the university scientific community. But changes must be made to improve it.

In the introduction section, it is necessary to remove issues that are related to the Covid pandemic (vaccines, self-medication, medicinal plants, telemedicine), but not related to the topic of the study (technostress). However, the objective is clearly stated. The reference to the fact that these are university medical students is missing.

Theoretical framework is correct, simple, brief and clear, but the aim of the work is included again. It can be removed from this section or from the previous one (introduction).

The approach lacks any reference to authors on the model used. The title of the section should be model (the representation scheme of the work that has been developed) and not focus (theory). The statement of the hypotheses is correct.

In the methodology section, the authors repeat again the aim of the study. The section is well exposed, both the methodology, the sample, instruments, variables and the data analysis procedure. However, the information on the origin of the instrument, its adaptation, the validation process by expert judgment, etc., must be completed.

Results are correctly presented, both the characteristics of the instrument (reliability and validity of the scale and subscales) used for data collection and the analysis of the informants' responses. The results achieved are correctly presented and commented on in detail, specifically regarding boostrapping, the proposed model and hypothesis testing. All sections are explained clearly and simply.

The discussion shows that it is necessary to specify who the university students are (only medicin students). Be careful, some comments are generalized to all students and the informants are medical students. These issues must be corrected. The first two paragraphs of this section are superfluous as they refer to general topics of the business environment, which are not part of the study.

Author Response

Reviewer #2

We thank the reviewer for the positive review of our manuscript. The reviewer has made some critical and insightful comments that have definitely improved the final version. We have carefully amended the paper as suggested by the reviewer.

Comments

It is an interesting article for the university scientific community. But changes must be made to improve it.

  1. In the introduction section, it is necessary to remove issues that are related to the Covid pandemic (vaccines, self-medication, medicinal plants, telemedicine), but not related to the topic of the study (technostress). However, the objective is clearly stated. The reference to the fact that these are university medical students is missing.

We thank the reviewer for this comment. However, we consider that is critical to mention briefly the impact that the COVID-19 pandemic had caused in Peru, specially considering that it points to the general state of disinformation in Peru and how people had to access online resources contributing to communication and social overload. We have added a sentence to point to this issue and have decided to maintain that introductory paragraph to help us position the situation in Peru.

  1. Theoretical framework is correct, simple, brief and clear, but the aim of the work is included again. It can be removed from this section or from the previous one (introduction).

We thank the reviewer for this comment, we have removed the aim from the Theoretical framework to avoid duplicity.

  1. The approach lacks any reference to authors on the model used. The title of the section should be model (the representation scheme of the work that has been developed) and not focus (theory). The statement of the hypotheses is correct.

We thank the reviewer for this comment, we have added the reference of the model used. The titles of the sections were based on the format of the journal.

  1. In the methodology section, the authors repeat again the aim of the study. The section is well exposed, both the methodology, the sample, instruments, variables and the data analysis procedure. However, the information on the origin of the instrument, its adaptation, the validation process by expert judgment, etc., must be completed.

We thank the reviewer for this comment, we have removed the aim from the Methodology to avoid duplicity. Related to the instrument, in section 4.2. Data collection and instrument we detail that we used instruments previously developed to assess communication overload, social overload, technostress, exhaustion and academic performance. The original items were translated to Spanish and adapted linguistically. Furthermore, the origins of the scales and items are presented in Table 1.

  1. Results are correctly presented, both the characteristics of the instrument (reliability and validity of the scale and subscales) used for data collection and the analysis of the informants' responses. The results achieved are correctly presented and commented on in detail, specifically regarding boostrapping, the proposed model and hypothesis testing. All sections are explained clearly and simply.

We thank the reviewer for the positive comments about our research.

  1. The discussion shows that it is necessary to specify who the university students are (only medicine students). Be careful, some comments are generalized to all students and the informants are medical students. These issues must be corrected. The first two paragraphs of this section are superfluous as they refer to general topics of the business environment, which are not part of the study.

We thank the reviewer for this comment, we have reduced the first two paragraphs of the Discussion section by removing the statements about the cell phone use, which did not correlate well with the other sentences. We have specified in the abstract and discussion section that our research surveyed university medical students. Furthermore, section 6.1. Limitations, indicate that: We surveyed university medical students, but technostress also needs to be evaluated in other professional careers and in other groups of people.